# Antagonism of BST2/Tetherin, a new restriction factor of respiratory syncytial virus, requires the viral NS1 protein

Katherine Marougka[1], Delphine Judith[2], Tristan Jaouen[1,3], Sabine Blouquit-Laye[1], Gina Cosentino[1], Clarisse Berlioz-Torrent[2]*, Marie-Anne Rameix-Welti[3,4☯]*, Delphine Sitterlin[1,3☯]*

1 Université Paris-Saclay, Université de Versailles St. Quentin, M3P, UMR 1173, INSERM, Versailles, France, 2 Université Paris Cité, Institut Cochin, INSERM, CNRS, Paris, France, 3 Institut Pasteur, Université Paris Cité, M3P, PARIS, France, 4 Université Paris-Saclay, Université de Versailles St. Quentin, UMR 1173, INSERM, Assistance Publique des Hôpitaux de Paris, Hôpital Ambroise Paré, Laboratoire de Microbiologie, DMU15 Versailles, France

☯ These authors contributed equally to this work.
* clarisse.berlioz@inserm.fr (CB-T); marie-anne.rameix-welti@pasteur.fr (MR-W); delphine.sitterlin@uvsq.fr (DS)

## Abstract

Human respiratory syncytial virus (RSV) is an enveloped RNA virus and the leading viral agent responsible for severe pediatric respiratory infections worldwide. Identification of cellular factors able to restrict viral infection is one of the key strategies used to design new drugs against infection. Here, we report for the first time that the cellular protein BST2/Tetherin (a widely known host antiviral molecule) behaves as a restriction factor of RSV infection. We showed that BST2 silencing resulted in a significant rise in viral production during multi-cycle infection, suggesting an inhibitory role during the late steps of RSV's multiplication cycle. Conversely, BST2 overexpression resulted in the decrease of the viral production. Furthermore, BST2 was found associated with envelope proteins and co-localized with viral filaments, suggesting that BST2 tethers RSV particles. Interestingly, RSV naturally downregulates cell surface and global BST2 expression, possibly through a mechanism dependent on ubiquitin. RSV's ability to enhance BST2 degradation was also validated in a model of differentiated cells infected by RSV. Additionally, we found that a virus deleted of NS1 is unable to downregulate BST2 and is significantly more susceptible to BST2 restriction compared to the wild type virus. Moreover, NS1 and BST2 interact in a co-immunoprecipitation experiment. Overall, our data support a model in which BST2 is a restriction factor against RSV infection and that the virus counteracts this effect by limiting the cellular factor's expression through a mechanism involving the viral protein NS1.

## Author summary

Respiratory syncytial virus (RSV) is the main cause of severe lower respiratory tract infection in children with more than 30 million cases worldwide every year. The innate

**Data Availability Statement:** All relevant data are within the manuscript and its Supporting Information files.

**Funding:** This work was supported by INSERM and Versailles Saint-Quentin University. MAW has received funding from ATIP-AVENIR INSERM program (2018) (https://www.inserm.fr/nous-connaitre/programme-atip-avenir/). MAW and SBL have received funding from the Fondation Del Duca - Institut de France (https://www.fondation-del-duca.fr/). KM and GC were supported by doctoral fellowship from Versailles St Quentin university. D. J. holds a fellowship from Agence Nationale de Recherches sur le Sida et les Hépatites Virales (ANRS; ECTZ60924) and then from SIDACTION (2021-2-FJC- 13113). The funders had no role in study design, data collection and analysis, decision to publish, or preparation of the manuscript.

**Competing interests:** The authors have declared that no competing interests exist.

immune response is a major determinant of RSV disease severity and drives the adaptive immune response. BST2 is a key protein of the innate immune response, which restricts several enveloped viruses by tethering the viral particles at the plasma membrane preventing viral release. In the present study, we demonstrate for the first time that RSV multiplication is attenuated by BST2 expression. We reveal that BST2 co-distributes with RSV viral particles at the plasma membrane, leading us to propose that BST2 behaves as a restriction factor by tethering RSV particles at the surface of infected cells. Interestingly, we report that RSV is able to downregulate cell surface BST2 expression. Finally, we discovered that RSV NS1 protein interacts with BST2 and participates to the mechanism by which RSV downregulates BST2. Moreover, the demonstration of RSV's counteraction to BST2 restriction in cells close to its natural targets underlines the relevance of these data and paves the way for new strategies of infection control.

## Introduction

RSV (Respiratory Syncytial Virus) or *human Orthopneumovirus* is a major global health concern. It is the leading infectious agent responsible for lower respiratory tract infections in young children worldwide. In addition, reinfections are frequent throughout life [1–4] and pose a threat to immunocompromised adults and the elderly. Recently, new antiviral approaches emerged for the treatment of RSV based on the use of the pre-fusion conformation of the viral F protein [5]. The search for antiviral drugs is still ongoing, especially to target steps other than viral entry or viral RNA synthesis [6]. Further understanding of the mechanisms involved in the viral cycle as well as of the interactions between the cellular host defenses and the virus remain necessary to develop new preventive and curative strategies.

RSV belongs to the *Pneumoviridae* family of the *Mononegavirales* order. RSV single negative strand genomic RNA of 15.2kb is tightly encapsidated by the Nucleoprotein (N) forming a helicoidal ribonucleocapsid [7]. Its enveloped virions are pleomorphic and range from spherical (150–250 nm in diameter) to filamentous that reach several micrometers in length (90–100 nm in diameter) [8–10]. On the surface of the viral particles, the G protein is involved in cellular attachment, while F drives fusion to the plasma membrane and subsequent entry in the cell [5]. Inside the particle, the ribonucleocapsids (N + viral RNA) are linked to the polymerase (L), its co-factor the Phosphoprotein (P) and the transcription factor M2-1 forming the viral ribonucleoproteins (vRNPs) [11,12]. *In vivo*, RSV is mostly targeting the ciliated bronchial epithelial cells [13] while *in vitro* it infects most cell lines. A large part of the viral cycle takes place into viral factories, which are formed in the cytoplasm by Liquid-Liquid-Phase-Separation (LLPS) to build an optimal environment for viral replication and transcription [14]. Then, newly synthesized vRNPs are suggested to reach the assembly sites at the plasma membrane by moving along the microtubules and by hijacking the endosome recycling pathway (mediated by Rab11) [15,16]. The Matrix protein (M) is mainly observed in contact with the plasma membrane where it is considered to be involved, along with the actin network, in the virion assembly and budding [17–20]. In addition to the proteins that are essential to its multiplication, RSV expresses non-structural proteins, NS1 and NS2. NS1 and NS2 are involved in type I and III interferon signaling pathways and prevent numerous important host antiviral factors [21].

BST2 (Bone Marrow Stromal Cell Antigen 2), also named Tetherin, is an interferon-stimulated type II transmembrane cellular glycoprotein of 180 amino acids localized at the lipid rafts of the plasma membrane, the trans-Golgi network (TGN) and the recycling endosomes

[22,23]. It is expressed in numerous tissues including respiratory cells [22]. BST2 is characterized by its unique topology that renders the attachment to membranes possible by both extremities *via* a short N-terminus cytoplasmic tail and a C-terminus glycosylphosphatidylinositol (GPI) anchor. These extremities frame an ectodomain with three cysteine residues that form disulfide bonds and thus drive the homodimerization of the protein. Mature BST2 is highly glycosylated due to the addition of N-glycans on two asparagine residues of the ectodomain [22,24]. BST2's antiviral functions were first described in 2008 on HIV-1 infection [25,26]. Briefly, BST2 restricts HIV-1 infection by "tethering" virions on the budding membrane thus restraining viral propagation [27,28]. BST2 also acts as a sensor of HIV-1 budding, leading to the establishment of a cellular anti-viral state [29,30]. However, this restriction is antagonized by HIV-1's accessory protein Vpu that orchestrates BST2 targeting to the endosomal pathway and excludes BST2 from the budding site [31–34]. Vpu also attenuates the NFkB-dependent pro-inflammatory response mediated by viral-induced BST2 signaling [30,35]. Since 2008, the antiviral effect of BST2/Tetherin was demonstrated for multiple enveloped viruses, notably including the *Filoviridae*, the *Paramyxoviridae* and the *Rhabdoviridae* families. Recently, BST2 was demonstrated to also disturb viral release of SARS-CoV-2 [36]. Different antagonistic mechanisms have been detected including but not limited to trapping BST2 in subcellular compartments, blocking its recycling or accelerating its degradation [37–39]. So far, the antiviral effect of BST2 has not been demonstrated for RSV but interestingly, secreted proteome analysis in a polarized pediatric airway epithelium model infected by RSV revealed a BST2 enrichment in the apical secretions of infected cells [40].

In the present study, we investigated whether BST2 restricts RSV infection and whether the virus antagonizes the antiviral effect of BST2. Initially, we used siRNA against BST2 and showed that the depletion of BST2 increased viral production, probably by enhancing the propagation of new virions. BST2 was found associated with envelope proteins and colocalized with viral filaments when overexpressed, further supporting the hypothesis that BST2 tethers RSV particles at the surface of infected cells. We revealed that RSV developed a strategy to counter the BST2 antiviral effect. Indeed, we observed that BST2 protein degradation was enhanced in infected cells, leading to a decrease of BST2 cell surface expression. RSV's ability to degrade BST2 was also validated in differentiated airway epithelium basal cells (BCi NS1.1). Interestingly, we showed that NS1 is required for BST2 antagonism during the course of RSV infection. We showed that BST2 interacts with NS1 and that RSVΔNS1, unable to encode NS1, is more sensitive to BST2 restriction than the wildtype virus as it fails to decrease BST2 cell surface expression.

## Results

### BST2 prevents RSV multiplication

To determine whether the role of BST2 as a restriction factor of some enveloped viruses would be extended to RSV, we first assessed the effect of BST2 depletion on RSV multiplication. Cells were thus transfected with control (si NT) or BST2 siRNA and kinetics of viral production were established by plaque titration assay 2 days later. In parallel, the depletion of BST2 at 2 days post transfection was controlled by Western blot (Fig 1A). As BST2 is known to prevent infection by tethering viral particles [27], we hypothesized that multicycle infection would be a sensitive strategy to reveal a BST2 inhibitory effect. We thus performed a kinetic at low MOI (multiplicity of infection) until 48h post infection. Interestingly, at 48h post infection we observed almost half-log difference in viral production between cells treated with BST2 siRNA and cells treated with control siRNA (Fig 1B). It should be noted that this difference is modest but highly reproducible. We did not observe any effect until 36h post infection because

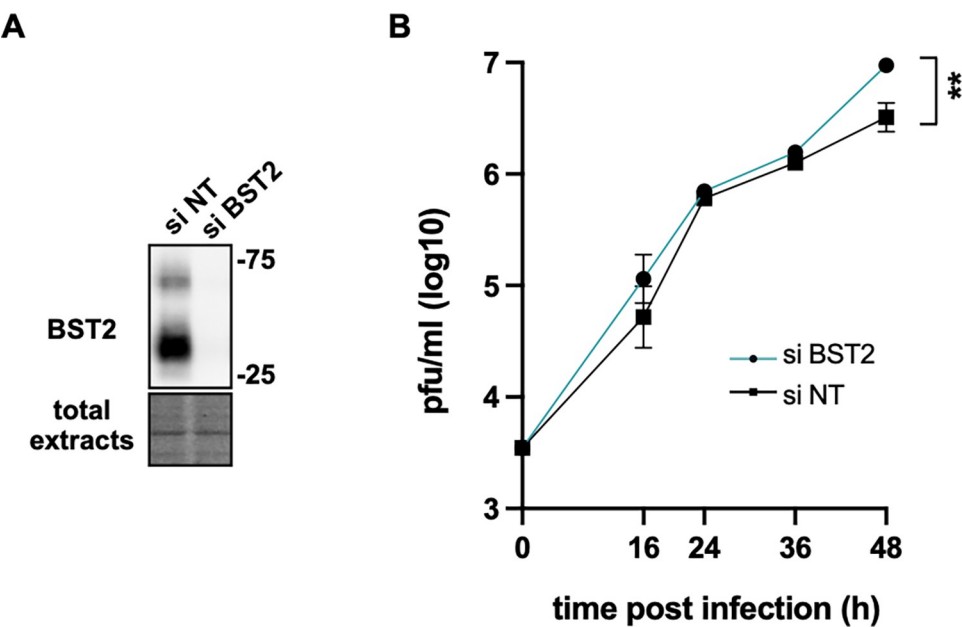

**Fig 1. Kinetics of viral infection on BST2 knockdown cells.** HEp–2 cells were transfected with control siRNA (NT) or siRNA targeting BST2. (A) Expression of BST2 was controlled by Western blot on total protein extracts at 48h post transfection. Below, an image of the total protein quantity loaded per well is presented. (B) Forty-eight hours post transfection, cells were infected with RSV WT at low MOI (0.01). At the indicated times post infection, viral titers were measured by plaque titration assay. Triplicates were realized for each time point per condition. Non-parametric unpaired $t$-tests were performed, ** $p < 0.01$. Representative data of 3 independent experiments are shown.

titration assays were performed on free and cell-attached viruses, since it is known that RSV viral particles remain attached to cell membranes in HEp–2 cells [8,10,17]. Consequently, the tethering effect could be hidden after one cycle of infection. Moreover, we showed that expression of viral proteins was not affected by BST2 siRNA treatment in a single round infection experiment performed at high MOI (S1 Fig), validating that early steps of the viral cycle were not disturbed by BST2.

To confirm the antiviral activity of BST2 under overexpression conditions, we first established HEp-2 cells containing a replicating episomal plasmid which overexpressed HA-tagged BST2 (HA-BST2) or HA as a control. Then cells were infected with a recombinant RSV virus expressing luciferase (RSV-Luc) as a reporter gene and a luciferase assay was performed 48h post infection to quantify the effect of BST2 overexpression in a multicycle infection assay. We showed that BST2 overexpression significantly affected RSV multiplication (S2 Fig). Altogether, our data indicate that BST2 affects the late steps of RSV multiplication and suggested that BST2 could act as a restriction factor of RSV infection.

## BST2 is associated with viral particles in infected cells

To further explore the hypothesis that BST2 tethers viral particles, we performed an immuno-precipitation experiment of endogenous BST2 to probe the interaction between viral particle components and BST2. HEp-2 cells were infected with RSV and 24h p.i. BST2 immunoprecipitation was realized on cell lysates. Total lysate and bound fractions were analyzed using an RSV antibody recognizing all viral structural proteins. As shown on Fig 2A, the G envelope protein was clearly identified in the bound fraction, as well as the N protein, although N was slightly present in the control immunoprecipitation. Using specific antibodies against G and F

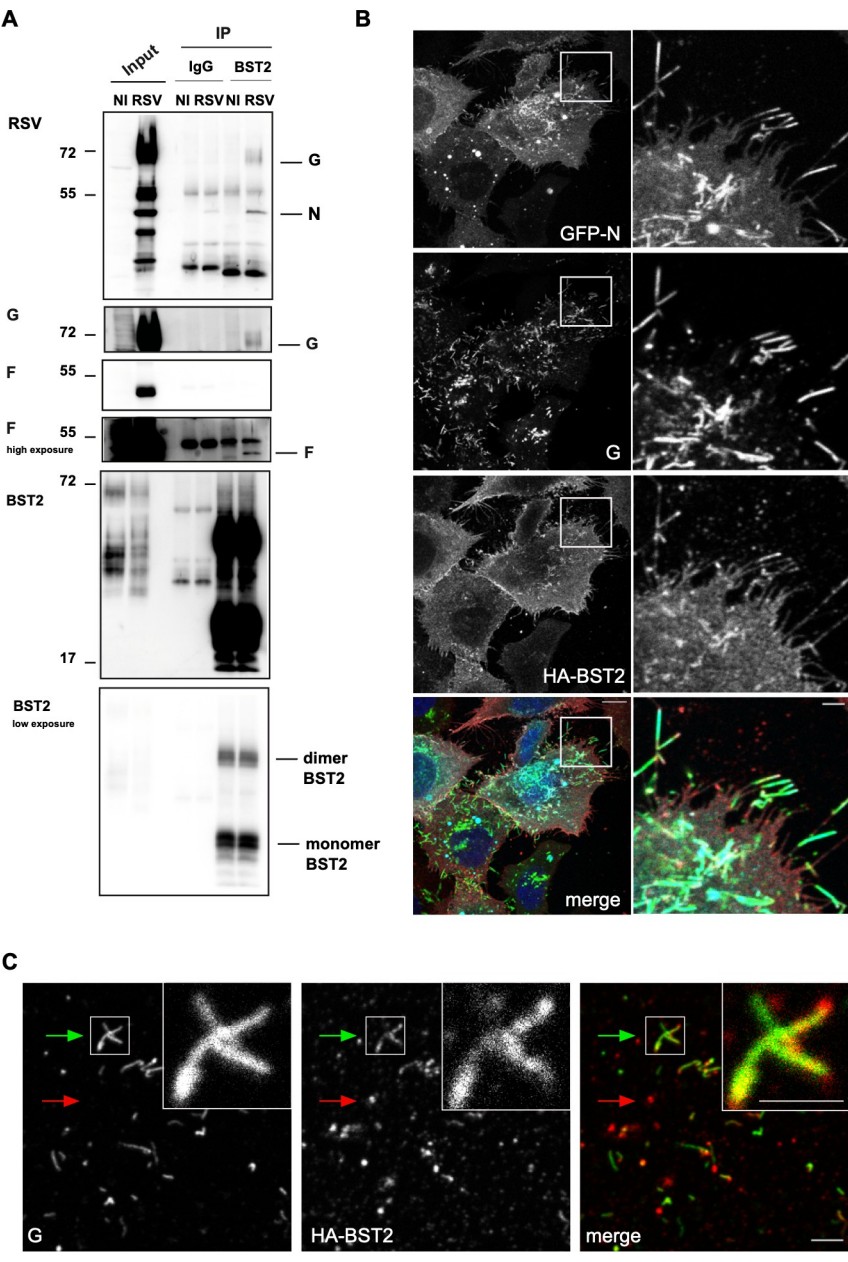

**Fig 2. BST2 association with viral filaments.** (A) HEp-2 cells were infected with RSV for 24h at MOI 1, and processed for IP using BST2 or a control antibody (IgG). IP samples were pretreated with PNGase before loading on the gel to improve visualization of the protein. Then input and IP were analyzed by WB using indicated antibodies on the left (B) HEp–2 HA-BST2 cells were infected with RSV-GFP-N for 24h at MOI 1. BST2 (red in merge) and G (green in merge) were revealed by immunostaining. The GFP-N protein is visualized through its spontaneous fluorescence (cyan in merge). Hoechst 33342 staining is shown in blue (merge). Representative images from 3 independent experiments are shown. One representative Z step is shown. Scale bar 10 μm (left panel). Zoom images of the white square are shown on the right. Scale bar 2 μm (right panel). (C) HEp–2 HA-BST2 cells were infected with RSV for 24h and viral particles were purified through sucrose cushion, fixed and stained with antibodies against BST2 (red in merge) and G (green in merge). One representative viral particle was shown with a green arrow, and hypothetical extracellular vesicle with a red arrow. Scale bar 3 μm. Enlarged images of the white square are shown on the upper right. Scale bar 2 μm.

envelope proteins, we confirmed the presence of G and showed the presence of F in the BST2 immunoprecipitation fraction. This result demonstrates an interaction between BST2 and viral envelope proteins, as well as with a component of the RNPs, the N protein, suggesting a possible interaction with viral particles. This result led us to analyze the distribution of BST2 at the vicinity of budding viral particles. This experiment was performed in HEp-2 cells overexpressing HA-BST2, which were infected with an RSV expressing a GFP-N fusion protein (RSV-GFP-N) [15], and processed for immunofluorescence at 24 hours p.i.. As shown on Fig 2B, when imaging the basal section of the cells with markers against G and N structural viral proteins, we observed BST2 in cell membrane extensions, colocalizing with G and N positive viral filaments. In contrast, HA-tagged TfR, an irrelevant membrane protein, overexpressed in the same conditions, was not found in the viral filaments (S3 Fig). To confirm the presence of BST2 in viral particles, we purified RSV particles produced from cells overexpressing HA-BST2 on a sucrose cushion and processed them for IF using anti-BST2 and anti-G antibodies (Fig 2C). We clearly observed BST2 associated with viral particles, as well as BST2 signal in spherical negative viral structures, likely extracellular secreted vesicles.

Altogether, these results suggest that BST2 is associated with RSV particles, notably when it is overexpressed. BST2 could be attached to one single particle or most likely could tether two or more particles together, as it occurs between plasma membrane and viral envelope. These results strengthen the proposed role of BST2 as a negative regulator of the late steps of RSV's life cycle.

## BST2 plasma membrane expression is decreased in RSV infected cells

We next analyzed the level of BST2 expression at the surface of infected HEp-2 cells. We used the previously characterized RSV-mCherry virus [41] to identify infected HEp–2 cells and determined by flow cytometry the level of BST2 present at the cell surface (Figs 3A and S4). Whereas HEp–2 cells expressed BST2 under basic growth culture conditions, we observed that the presence of BST2 was significantly reduced (30% decrease) at the cell surface of RSV-mCherry infected cells compared to non-infected cells. As a control, we analyzed MHCI distribution at the plasma membrane in infected and non-infected cells and we did not observe any difference between the conditions (S5 Fig). We confirmed this result by imaging BST2 in RSV infected cells at 24h post infection. To that end, immunofluorescence was performed on infected cells using BST2 antibody together with N (marker of infected cells) and TGN46 (marker of the Trans Golgi Network) antibodies. As displayed on Fig 3B, we clearly showed the absence of BST2 signal at the cell surface in RSV-infected cells, while the TGN signal was still observed in both conditions. These observations were quantified in control and RSV-mCherry infected cells using phalloidin and GM130 markers to reconstruct in 3D the cellular plasma membrane and the TGN network, respectively (Fig 3C, Imaris Analysis software). The percent of BST2 remaining at the surface was calculated as the ratio between BST2 plasma membrane intensity signal and BST2 "total" intensity signal ("total" representing BST2 plasma membrane intensity signal + BST2 signal found in TGN). We clearly showed a significant decrease of the percent of BST2 present at the plasma membrane in infected cells compared to non-infected cells (Fig 3C). This result is consistent with the flow cytometry data.

We next analyzed whether the decrease of BST2 at the cell surface is due to a global decrease of BST2 expression. We first evaluated the mRNA level in infected cells by RT-qPCR analysis. We showed that the BST2 mRNA level was not significantly modified in infected cells compared to non-infected cells, suggesting that the regulation of the amount of BST2 doesn't take place at the transcriptional level (Fig 4A and S1 Table). We then performed Western blot on total protein extract at 24h p.i. to evaluate BST2 protein level in the infected cells. As shown on Fig 4B, we clearly observed a large decrease of the total amount of BST2 in infected cells

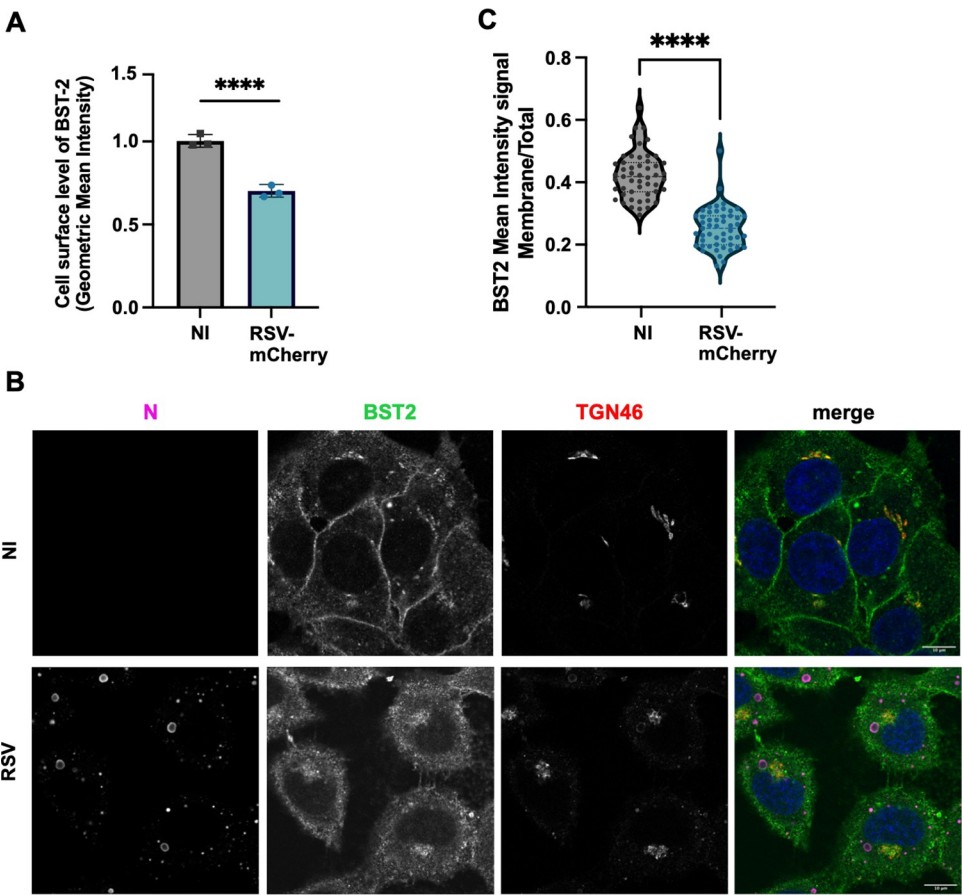

**Fig 3. BST2 localization during RSV infection.** (A) HEp–2 cells were infected or not with RSV-mCherry. At 22h p.i. cells were fixed and stained with an antibody against BST2. The Geometric Mean Intensity (GMI) of BST2 at the surface of the cells (without permeabilization) was determined by Flow Cytometry in the two populations. (B) HEp–2 cells were infected or not with RSV. At 24h p.i. cells were fixed and stained with antibodies against TGN46 (red in merge), N (magenta in merge), and BST2 (green in merge). Hoechst 33342 staining is shown in blue (merge). Representative images from 3 independent experiments are shown. One representative Z step is shown. NI (not infected). Scale bar 10 μm. (C) HEp-2 cells were infected or not with RSV-mCherry. At 24h p.i. cells were fixed and stained for actin (phalloidin), BST2 and GM130 (cis-golgi marker). Quantification of BST2 signal was performed using the Imaris Software as described in methods section. The BST2 mean signal at the plasma membrane was normalized to the total signal detected in the cells (plasma membrane + TGN). Representative data of 2 independent experiments. In each experiment, 30 cells per condition were analyzed. (A, C) Non-parametric unpaired *t*-tests were performed on the Prism 9.5.0 Software, **** p< 0.0001.

compared to non-infected cells (2.4-fold, quantification on S6 Fig), leading us to hypothesize that the degradation of BST2 is enhanced in infected cells. To confirm this observation, we further investigated the degradation route of BST2. RSV infected cells were treated either with MG-132, an inhibitor of the proteasome pathway or bafilomycin A1, an inhibitor of the endo-lysosomal degradation pathway. As shown on Fig 4C and 4D (quantification S7 and S8 Figs), both treatments blocked BST2 degradation significantly. We next examined the level of BST2 ubiquitination in RSV infected cells by blotting immunoprecipitated BST2 with anti-ubiquitin antibodies. Interestingly, we observed higher amounts of ubiquitinated forms of BST2 in RSV infected cells that in the non-infected condition (Fig 4E), suggesting that RSV infection increases BST2 ubiquitination to favor its degradation. These results suggested that the degradation of the BST2 protein upon infection could be dependent on both endolysosomal and ubiquitin pathways. Additionally, immunofluorescence experiments were performed with

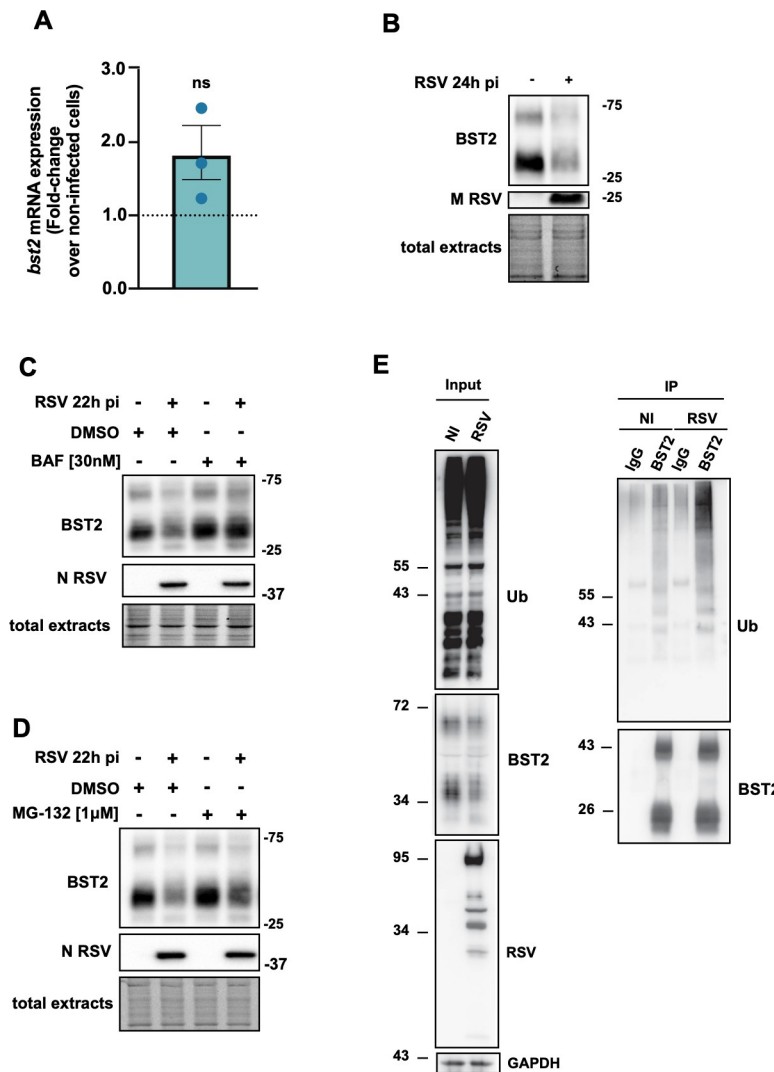

**Fig 4. RSV infection enhances BST2 ubiquitination and degradation.** (A-D) HEp–2 cells were infected or not with RSV. (A) At 22h p.i. total RNA was extracted from the cells. Reverse transcription and qPCR were performed. Data were analyzed using the $2^{-\Delta\Delta Ct}$ method. Actin B mRNA was used to normalize the mRNA quantity between conditions. One sample t and Wilcoxon test were realized (Ct value S1 Table). (B) At 24h p.i. total protein extracts were obtained and processed for Western blot. Membranes were incubated with antibodies against BST2 and viral protein M. (C-D) At 14h p.i. cells were treated with DMSO, Bafilomycin A1 [30nM] or MG-132 [1µM]. At 22h p.i., total protein extracts were obtained and processed for Western Blot using BST2 and N antibodies. For all Western Blot, an image of the total protein quantity deposited per well is presented, and quantifications were shown on S5–S7 Figs. Representative results of 3 independent experiments. (E) HEp-2 cells were infected for 24h. Endogenous BST2 was immunoprecipitated either with anti- IgG control or mouse anti-BST2 antibodies (BST2 M15). IP samples were pretreated with PNGase before loading on the gel to improve visualization of the protein. Then input and IP were analyzed by Western blot using anti-FK2-HRP and rabbit anti-BST2 antibodies.

CD63 (a late endosome marker) antibodies, together with TGN46 and showcased that upon infection, BST2 is downregulated and mainly present in TGN46 positive compartments while less present in CD63 compartments (S9 Fig).

Altogether, our data suggested that BST2 ubiquitination and then degradation are enhanced in infected HEp-2 cells, leading to a reduced exposure of BST2 at the plasma membrane.

## BST2 is downregulated in a physiological airway epithelial cell culture model

We investigated BST2 behavior in a bronchial epithelial cell line (BCi-NS1.1, Basal Cells immortalized-NonSmoker 1.1) that more closely resembles RSV's natural targets [42]. BCi-NS1.1 is an epithelial cell line derived from human basal cells, that differentiates into a polarized epithelium with ciliated cells and secretory cells when cultured at an air-liquid interface on permeable supports. These cells have previously been shown to be susceptible to RSV infection [43]. Differentiated BCi-NS1.1 cells were infected with RSV-GFP for 72 hours p.i. and BST2 expression was assessed by western blot on whole cell extracts. BST2 protein is not detected in non-infected cells but is expressed at 3 days post-infection (Fig 5A). This result was expected as it has been shown in primary airway epithelial cells that BST2 belongs to a core of

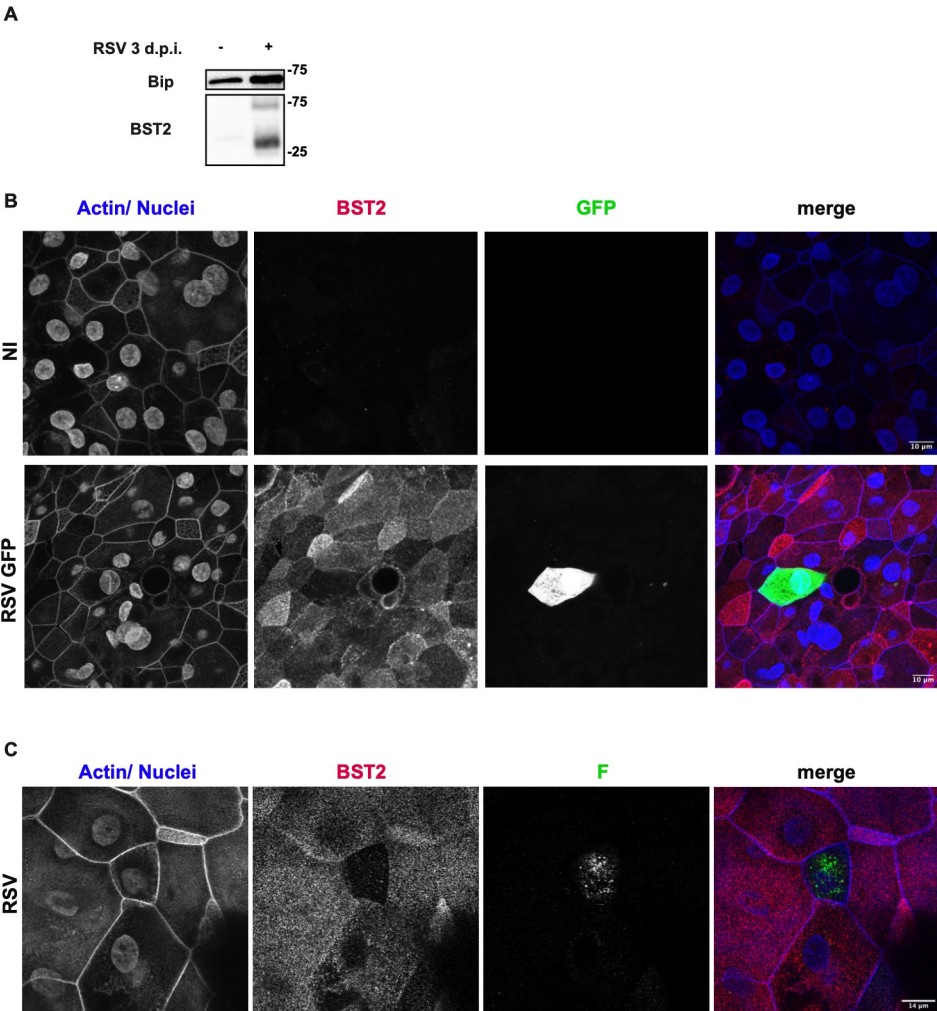

**Fig 5. Expression of BST2 in differentiated epithelial cells infected with RSV.** BCi-NS1.1 cells after 3 weeks in ALI culture were infected or not with RSV (A, C) or RSV-GFP (B) at high (1) MOI for 3 days. (A) Total protein extracts were processed for WB. Binding immunoglobulin protein (Bip) was used as gel loading control. (B) Cells were fixed and stained for immunofluorescence to reveal actin and nuclei (in blue) and BST2 (in red). The infected cells are expressing the GFP (in green). (C) Cells were fixed and stained for immunofluorescence to reveal actin and nuclei (in blue), BST2 (in red) and F viral protein (green). Confocal image stacks (14 z-steps) were processed as maximum projections. Representative results out of two independent experiments.

IFN stimulated genes activated upon RSV infection [44]. BST2 localization was also assessed by immunofluorescence in the infected epithelia (Fig 5B). In the infected sample, BST2 was detected in all cells thus confirming BST2 expression induction by RSV infection. Interestingly in the specific cells replicating the virus, we do not observe BST2 signal. This result was confirmed using a non-fluorescent RSV virus (Fig 5C, infected cells were detected with a F staining). Altogether, these results suggest that the virus also managed to counteract BST2 in a more physiological model of infection.

## RSV-induced BST2 downregulation requires the expression of viral NS1 protein

We next investigated which viral protein is responsible for BST2 downregulation in RSV infected cells. We previously showed on Fig 2A an interaction between RSV envelope proteins and BST2, leading to the hypothesis that G or F could be one of these proteins. To explore this hypothesis, an RSVΔG virus was rescued by reverse genetics. HEp–2 cells were infected with RSVΔG or RSV WT viruses for 24h and processed for IF in order to evaluate BST2 membrane level. As BST2 was still downregulated following infection with RSVΔG (S10 Fig), we concluded that G is not the viral protein responsible of BST2 antagonism. Unfortunately, it was not possible to address a similar question for F as an RSVΔF virus is not sustainable.

BST2 could most likely be antagonized by RSV NS (NS1 or NS2) proteins which are highly efficient host antiviral response regulators. First, we probed a possible interaction between BST2 and NS proteins. HEp–2 cells were transiently transfected with expression plasmids of Flag-tagged NS1 (Flag-NS1), Flag-tagged NS2 (Flag-NS2) or HA-tagged NS2 (HA-NS2) and Flag-NS1 genes. Then, immunoprecipitation of endogenous BST2 was performed on cell lysates. The presence of NS1, NS2 and BST2 in the cell lysates and the bound fraction was analyzed by Western Blot (Fig 6). Interestingly, we found that NS1 protein and, less efficiently, NS2 protein immunoprecipitated with BST2. However, when NS1 and NS2 were expressed together, NS1 protein but not NS2 was immunoprecipitated with BST2.

Consequently, we focused on the ability of NS1 to downregulate BST2 in a viral context. We explored whether an RSV-mCherry virus that does not express the NS1 gene (RSV-mCherryΔNS1) is still able to downregulate BST2 at the surface. RSV-mCherryΔNS1 virus was rescued by reverse genetics and HEp–2 cells were infected with RSV-mCherryΔNS1 or RSV-mCherry WT viruses for 24h. Cells were then processed for immunofluorescence to localize BST2 in infected cells and most particularly at the plasma membrane as in Fig 3B and 3C. Remarkably, we showed that BST2 is no longer decreased at the plasma membrane of cells infected with the RSV-mCherry ΔNS1 virus, compared to cells infected with WT RSV-mCherry (Fig 7A and 7B). To further demonstrate that NS1 is able to affect BST2 mediated restriction, we assumed that the RSV-mCherry ΔNS1 virus should be more sensitive to BST2 restriction than the wild type virus. Consistent with the results of Fig 1, we performed RSV-mCherry ΔNS1 or RSV-mCherry infection on BST2 or Control silenced cells at low MOI and measured viral production 48h post infection by plaque titration assay. In agreement with the literature [45], the replication efficiency of these two viruses was different. We thus performed a supplementary assay for RSV-mCherry ΔNS1 at 72 h post infection. Results were expressed as viral titer fold change between BST2 silenced cells and control silenced cells for each virus (Fig 7C and S2 Table). We clearly showed that RSV-mCherry ΔNS1 is significantly much more sensitive to BST2 restriction compared to RSV-mCherry at 48h and 72h post infection. NS1 is known to have a broader anti-IFN role leading to a decrease of IFN stimulated genes at a transcriptional level, we thus checked whether our last observation could be explained by higher expression of BST2 mRNA in RSV-mCherry ΔNS1 infected cells. To investigate this

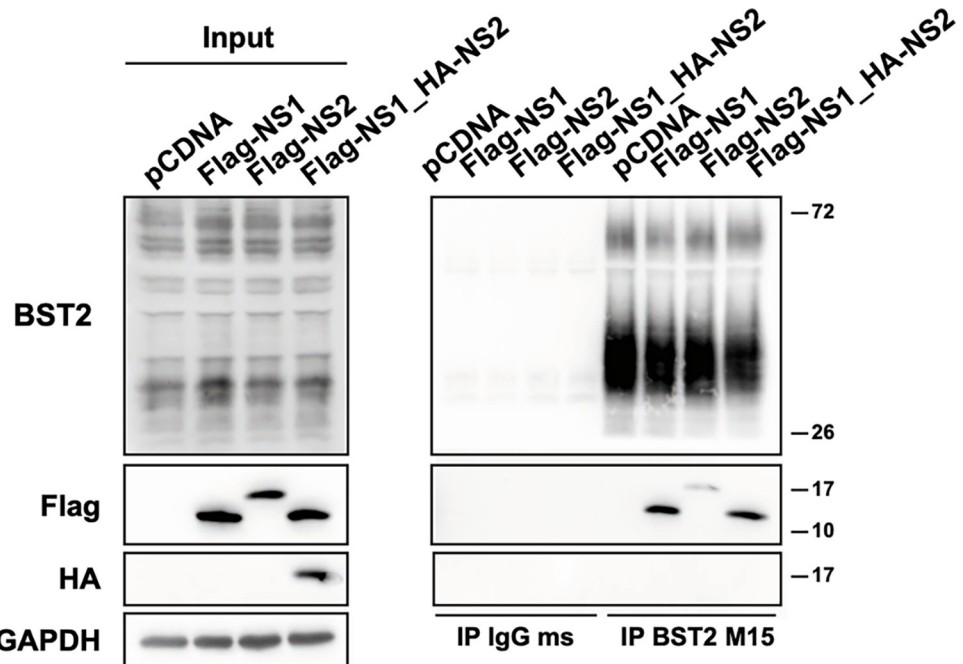

**Fig 6. BST2/NS1 interaction.** HEp-2 cells were transfected with the indicated plasmids for 24h. Cell lysates were immunoprecipitated either with anti-IgG control or mouse anti-BST2 antibodies (BST2 M15) and immunoprecipitated proteins were detected by western blot using rabbit anti-BST2, anti-Flag-HRP and anti-HA-HRP antibodies.

hypothesis, we quantified by qRT-PCR BST2 mRNA expression in cells infected by the two viruses. The results clearly showed that the BST2 mRNA expression was not modified in RSV-mCherry ΔNS1 compared to RSV-mCherry infected cells, whereas IFN-β, used as a positive control, was significantly more expressed in RSV-mCherry ΔNS1 infected cells than in RSV-mCherry infected cells (Fig 7D and S1 Table). This result allowed us to conclude that the effect of NS1 on BST2 expression is specific and not related to its broader role on the interferon pathway.

Finally, we investigated the ability of NS1 to rescue the release of retroviral particles from restriction by BST2. For that, we analyzed the release of a mutated HIV-1 virus that does not express Vpu, the HIV-1 antagonist of BST2, in cells transfected with Vpu-deleted HIV-1 provirus (HIV-1 Udel) and either NS1 or NS2 or NS1 + NS2. As shown in Fig 8, neither NS1 nor NS2 nor NS1 in combination with NS2 are able to counteract BST2 restriction on HIV-1 release, indicating that NS1 alone or in complex with NS2 are not sufficient to antagonize BST2 antiviral effect on HIV-1 release.

Altogether, these results demonstrate that NS1 is required, but not sufficient, to antagonize BST2 upon RSV infection. Indeed, we showed that *i)* the two proteins interact in a co-immunoprecipitation experiment and *ii)* an RSVΔNS1 virus is more sensitive to BST2 restriction than the wild type virus and fails to decrease BST2 cell surface expression. However, NS1 alone or in complex with NS2 is not sufficient to antagonize BST2 restriction on retroviral release, suggesting that NS1 requires additional viral proteins to counteract BST2 restriction.

## Discussion

BST2 (also named Tetherin) is known to be a restriction factor of a large range of enveloped viruses that bud at the plasma membrane, by preventing the release of viral particles from

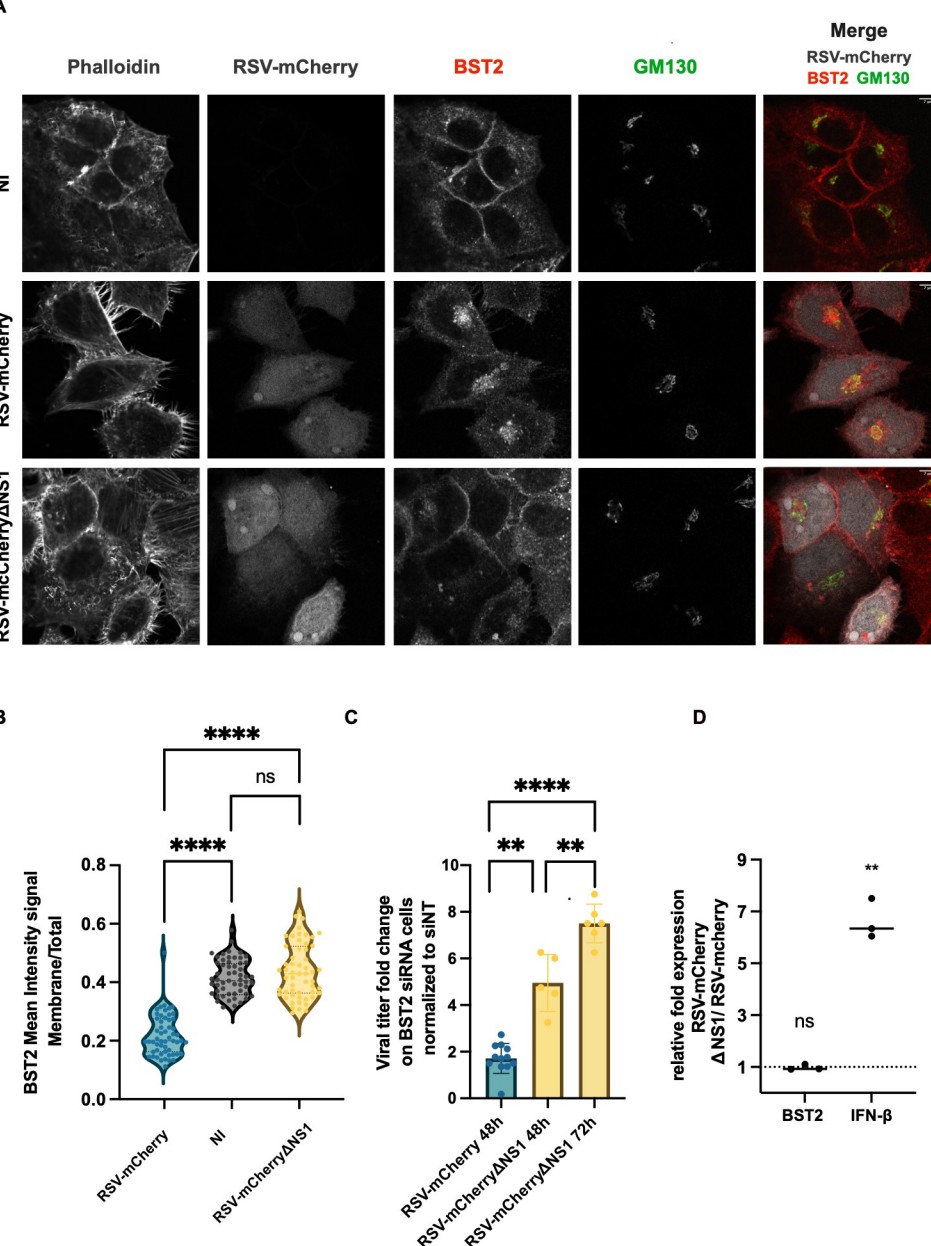

**Fig 7. BST2 restriction impaired in RSV-mCherryΔNS1 infected cells.** (A) HEp–2 cells were infected or not with RSV-mCherry or RSV-mCherryΔNS1. At 24h p.i. cells were fixed and stained against actin (phalloidin), BST2 (red in merge) and GM130 (cis-golgi marker, green in merge). The mCherry protein is visualized through its spontaneous red fluorescence (grey in merge). Representative images from 3 independent experiments are shown. One representative Z step is shown. NI (not infected). Scale bar 8 μm. (B). Quantification of BST2 signal was performed using the Imaris Software as described in methods section. The BST2 mean signal at the plasma membrane was normalized to the total signal detected in the cells (plasma membrane + TGN). Representative data of 3 independent experiments. In each experiment, 30 cells per condition were analyzed. Non-parametric unpaired $t$-tests were performed on the Prism 9.5.0 Software, **** p< 0.0001. (C) HEp–2 cells were transfected with control siRNA (NT) or siRNA targeting BST2. Forty-eight hours post transfection, cells were infected with RSV-mCherry or RSV-mCherryΔNS1 at low MOI (0.01). At 48h post infection (and 72h for RSV-mCherryΔNS1), viral titers were measured by plaque titration assay. Viral titer fold changes between siRNA BST2 and NT HEp-2 cells were represented for both viruses (average of viral production for each point were provided on S2 Table). Non-parametric unpaired $t$-tests were performed, ** p< 0.01. (D) Comparison of BST2 mRNA level in RSV-mCherryΔNS1 and RSV-mCherry infected cells. HEp–2 cells were infected with RSV-mCherryΔNS1 and RSV-mCherry. At 22h p.i. total RNA was extracted from the cells. Reverse transcription and qPCR were performed to amplify BST2 or IFN-β. GAPDH mRNA was used to normalize the mRNA quantity between conditions. Data were analyzed using the $2^{-\Delta\Delta Ct}$ method. One sample t and Wilcoxon test were realized.

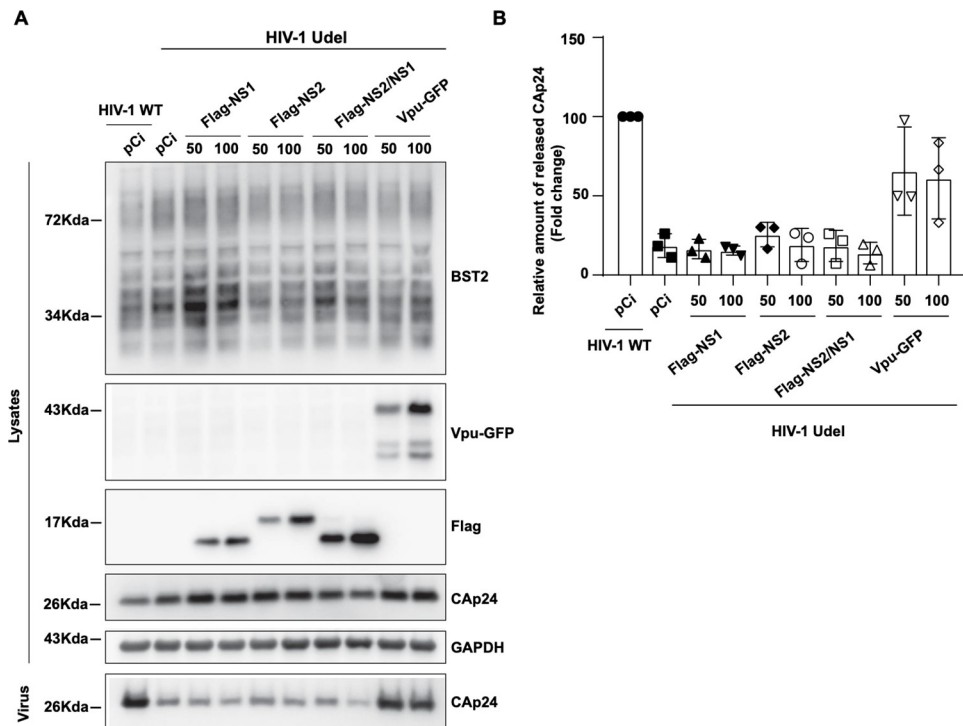

**Fig 8. NS1 does not antagonize BST2's inhibitory effect on HIV-1 release.** HeLa cells were co-transfected with provirus HIV-1 WT or Vpu-deleted (Udel) in combination with empty vector (pCi) or expression vector encoding Flag-NS1, Flag-NS2 or Vpu-GFP. After 48 hours of transfection cell lysates and pelleted supernatants were analyzed for HIV-1 CAp24 by western blot (A) and ELISA (B). Cell lysates were also analyzed for Flag-NS1, Flag-NS2, Vpu-GFP and GAPDH by western blot. Western blots are representative of at least three independent experiments (n = 3 experiments).

infected cells. Most of these viruses have developed antagonistic strategies that counteract tetherin's function, ensured by select viral proteins. In this paper, we enlarged its target spectrum, as we proposed for the first time that BST2 could also restrict the production of the respiratory syncytial virus. Indeed, we demonstrated that RSV multiplication is affected by BST2 and showcased evidence suggesting that the RSV non-structural protein NS1 is required to antagonize BST2.

To demonstrate that BST2 behaves as a restriction factor, we abolished its expression using siRNA in HEp–2 cells. The depletion of BST2 leads to an improvement of RSV viral production 48h post infection (half-log difference); the limited effect observed could be explained by the natural downregulation of BST2 by the virus which occurs even in the control cells. The restriction effect of BST2 was confirmed using cells constitutively overexpressing BST2 through a replicating episomal plasmid. It should be noted that in 2020, Li et al. showed in a control experiment that BST2 did not restrict RSV replication by measuring viral production in a multicycle assay [46]. This study used HEK293T cells, which did not express BST2 in physiological conditions, and BST2 restriction was addressed by overexpressing BST2 through transient transfection. In our study, we used HEp–2 cells, commonly used for RSV infection which expressed BST2 in physiological conditions, preventing uncontrolled side effects on the efficacy of RSV infection due to BST2 overexpression or due to viral countermeasures against a restriction factor.

Few restriction factors are known to sway RSV infection. Most of them affect host cell binding (CXCL4), replication/ transcription steps (IFITM, IFI44, A3G), or production of defective

viral particles (GBP5) [46–51]. Remarkably, to our knowledge, this study is the first to identify a cellular restriction factor affecting the late steps of RSV viral cycle. Our results are in agreement with the literature as BST2 is known to restrict the release of viral particles through a tethering mechanism [26]. We demonstrated that overexpressed BST2 is co-localizing with viral filaments (with G and N proteins). Moreover, we found that components of the viral particles are immunoprecipitated together with BST2 (F, G and N proteins). Despite the difficulty to gauge the tethering effect of BST2 in our HEp-2 cellular model, due to RSV viral particles remaining attached to cell membranes [8,10,17], our data support a model in which BST2 interacts with RSV's envelope by its transmembrane domain or its GPI anchor, preventing release of viral particles by tethering several secreted viral particles together.

In a model of HEp–2 cells where the BST2 gene is expressed in normal growth conditions, we observed a downregulation of BST2 cell surface expression under infection. This decrease is linked to a strong reduction of the total level of the protein, which is prevented by lysosomal inhibitor Bafilomycin A1 and, to a lesser extent, by proteasomal inhibitor, MG-132. We also showed an increase of ubiquitinated BST2 during infection and a decrease of BST2 presence in CD63 compartments compared to non-infected cells. We hypothesized that the increased ubiquitinylation of BST2 during infection accelerated its degradation. Alternatively, as we observed an accumulation of BST2 in the TGN, we could postulate that the recycling of BST2 to the cell surface could be altered in response to infection, rerouting BST2 to the TGN, leading to its degradation.

BST2 downregulation is supported by the use of the BCi-NS1.1 (Basal Cells immortalized-NonSmoker 1.1) cells, a more physiological model for RSV infection, which do not synthesize BST2 in normal growth conditions. Indeed, in BCi-NS1.1 infected cells, we clearly observed a strong downregulation of BST2 expression by immunofluorescence, whereas the induction of BST2 expression, due to IFN response to viral infection is clearly observed in neighboring uninfected cells (Fig 5). As the percent of infected cells is low in this cell line, Western blot results only display the upregulation of BST2 due to IFN response. Using two distinct cellular models with different basal levels of BST2, we documented a similar RSV-induced BST2 down regulation, demonstrating that the virus could counteract the restriction factor BST2.

Studying the numerous targets of BST2 in enveloped viruses is often followed with the discovery of viral proteins that interact with the restriction factor and counteract its antiviral effect. As mentioned, Vpu of HIV-1 successfully antagonizes BST2 antiviral activities by excluding it from the budding site and limiting BST2 expression at the cell surface and by enhancing its degradation [25,26,33,34,39]. Frequently, envelope glycoproteins assume that role as is the case for EBOV's glycoprotein (GP) that overcomes the retention of EBOV VLPs at the cell surface [52]. Similarly, the gpM of HSV-1 diminishes cell-surface BST2 [53]. However, the restriction of CHIKV VLPs is hindered by its non-structural protein 1 (nsP1) though a mechanism is not defined yet [54]. Interestingly, KSHV encodes for K5 a RING-CH E3 ubiquitin ligase that favors BST2 ubiquitination and leads to endosomal degradation [55]. The first proof of evidence to demonstrate the role of an antagonist protein is to showcase the absence of downregulation in a knockout virus for this protein. Our data show that the RSV-mCherryΔNS1 virus does not downregulate BST2 signal (Fig 7B) and that the RSV-mCherryΔNS1 is much more sensitive to BST2 restriction (Fig 7C). It should be noted that this effect is specific and not related to the ability of NS1 that allows RSV to escape the host IFN antiviral response. Indeed, NS1 is localized in the cytoplasm where it interacts with MAVs or TRIM25, two sensors of the antiviral response, impairing their activity [56]. NS1 can also affect the IFN receptor signaling pathway, or interacts in the nucleus with the mediator complex to modulate its transcriptional capacity on IFN response genes [57–59]. It is well known that NS1 and the other non-structural protein NS2, cooperate to suppress IFN response, probably through

formation of a NS1/NS2 heterodimer [60]. In our data, NS2 is not co-immunoprecipitated with BST2 when it is co-expressed with NS1 (Fig 6), suggesting that NS1 action is independent of NS2. However, as NS2 alone is slightly immunoprecipitated with BST2, we could not completely exclude a role of NS2 and further experiments should be undertaken to analyze it. In our model, we propose that in infected cells, NS1 could interact with BST2, possibly leading to the recruitment of a E3 ubiquitin complex which could enhance BST2 ubiquitination and favor its targeting to the degradation. Indeed, it has been reported that NS1 contains an interaction domain with E3 ubiquitin ligase elongin C and cullin 2, meaning that this protein could be able to recruit E3 ubiquitin ligase in order to favor BST2 ubiquitination and trigger protein degradation [61].

The majority of our results were obtained using non-differentiated cells (HEp-2 cells, commonly used for RSV research), but the restriction of BST2 on RSV spread should be further addressed in primary BCi-NS1.1 cells in the future. Follow-up studies will also be undertaken to explore more closely the NS1/BST2 interaction, including engineering NS1 mutant viruses in the putative E3 ubiquitin ligase interaction domain, or assaying direct *in vitro* interaction between NS1 and BST2. Mutants of BST2 should also be carried out to study which domain is involved in the recruitment to the viral filamentous particles.

In summary, our data support a model in which BST2 is a new restriction factor of RSV. RSV is able to downregulate BST2 expression through a protein degradation mechanism dependent on ubiquitin. We showed that NS1 is involved in the mechanism by which RSV counteracts BST2 restriction. However, our experiments have been performed in a viral context and we cannot exclude that an additional viral protein could be also necessary to downregulate BST2. This hypothesis is supported by the fact that NS1 alone is unable to counteract BST2 restriction on HIV-1 particle release. The additional viral factor is not NS2 since addition of NS2 in combination with NS1 does not rescue viral release either. Additional experiments could be set up to explore the contribution of other RSV proteins acting with NS1 to antagonize BST2, such as the F or N proteins that we found associated to BST2 in RSV infected cells (Fig 2A and 2B) or the SH protein, a small transmembrane protein that resembles the viroporine-like HIV-1 Vpu. Although the molecular details of how NS1 recruits components necessary to BST2 degradation remain to be elucidated, our study brings to light a new mechanism of viral opposition to cellular host defenses.

## Materials and methods

### Cell culture

HEp–2 cells (ATCC, CCL-23) were maintained in Eagle's minimum essential medium (MEM), HeLa, BHK-21 cells (clone BSRT7/5) [62] and Vero cells (ATCC, CCL-81) in Dulbecco's modified essential medium (DMEM), supplemented with 10% heat-inactivated fetal bovine serum (FBS) and with 1% penicillin–streptomycin solution 5000 U/mL. The BCi-NS1.1 cell line was cultured on permeable supports at air-liquid interface conditions. In these conditions it develops a differentiated polarized and ciliated epithelium [42]. All cells were grown at 37˚C in 5% $CO_2$.

### Viruses and plasmids

HA-Tagged hTfR (hTfr-HA) and BST2 (HA-BST2) constructs have been provided by addgene (#69610) and Dr Katy Janvier [63]. TfR-HA was PCR amplified and cloned into a modified form of the pCEP4 mammalian expression vector (Thermo Fisher) previously characterized in [64], between the XhoI and KpnI restriction sites.

Flag-NS1 and NS2 codon optimized genes were a kind gift from Dr Monika Bajorek. Then Flag-NS1, Flag-NS2 and HA-NS2 were cloned as follows into the pCi mammalian vector

(Promega): Flag NS1 was cloned at the XhoI/XbaI sites and Flag-NS2/HA-NS2 at the EcoRI/NotI sites.

The genome of the recombinant viruses originated from the human RSV strain Long sequence (ATCC VR-26). Wild type RSV, RSV-mCherry, RSV-Luc, RSV-GFP, RSV-PBFP and RSV-GFP-N were described previously [15,41,65,66]. RSV-mCherryΔNS1 reverse genetic vector was obtained by replacing, in the RSV-mCherry vector, the region framed by the SmaI-KpnI sites (containing both NS genes) with the NS2 gene. The RSV-mCherryΔNS1 vector was used to produce the corresponding virus through reverse genetics as previously described [65]. The rescued virus was amplified in Vero cells. RNA was extracted from the viral stock using the QIAamp viral RNA extraction kit (Qiagen 221413). Viral genomic RNA was specifically amplified (SuperScript IV One-Step RT-PCR System with ezDNase, Thermo Fisher) and the expected sequence was confirmed by sequencing. Primer sequences are available upon request. The nucleotide sequence of RSV-mCherryΔNS1 was deposited in the Genbank nucleotide database (RSV-mCherry-deltaNS1, OQ689796 accession number).

## Treatments with degradation inhibitors

HEp–2 cells were infected with RSV at high MOI (1) and treated with degradation inhibitors at 14 hours post infection (p.i.) for 8 hours. Bafilomycin A1 (Sigma-Aldrich; 19–148) and MG-132 (Sigma-Aldrich; M8699) were used in a final concentration of 30 nM and 1 μM, respectively, diluted in Dimethyl sulfoxide (DMSO).

## Cell infection and transfection

Viral stock was diluted in MEM or DMEM media devoid of FBS and then used to infect the cells. After 2 hours of adsorption, the inoculum was replaced by media supplemented with 2% FBS. During BCi-NS1.1 infection, the inoculum was not removed. HEp–2 cells were transfected with non-targeting siRNA (Dharmacon D-001206-14-05) or siRNA targeting BST2 (5' GAAUCGCG GACAAGAAGUA 3'; Sigma) at 10 nM for 100.000 cells and seeded in p24 plates. RNAiMax Lipofectamine (Thermo Fisher) was used as transfection reagent following the manufacturer's recommendations. For plasmid transfection, ten million HEp–2 cells were transfected with either 10 μg of pCEP4m plasmid DNA or with 5 μg of Nonstructural proteins/empty vector using using Polyethylenimine (PEI) as the transfection reagent following the manufacturer's recommendations. For the cells transfected with the pCEP4m vector, 2 days post transfection the culture medium was replaced with MEM medium supplemented with 150 μg /mL Hygromycin (Invivogen). The selective medium was replaced every 2 days for 10 days to select cells containing the replicating episomal vector. The cells were maintained for 5 passages.

## Kinetics of virus production and plaque titration assay

Forty-eight hours after transfection, siRNA transfected HEp–2 cells were infected with RSV, RSV-mCherry or RSV-mCherryΔNS1 at low MOI (0.01). At different times p.i., the plates containing the cells were directly frozen at −80˚C. Upon thawing, virus production was analyzed by plaque titration assay as previously described [65]. For RSV-mCherry and RSV-mCherryΔNS1, the number of mCherry foci were counted using an epi-fluorescence microscope. Triplicates were realized for each time point per condition.

## RSV-Luc titration assay

HA-BST2 or HA HEp-2 cells were infected with RSV-Luc at MOI of 0,01 and 48 h post infection, cells were lysed in a lysis buffer (25 mM Tris pH 7.8, 8 mM MgCl2, 1% Triton X-100,

15% glycerol, 1 mM dithiothreitol) for 15 min at room temperature with gentle shaking. We used a Tecan infinite M200PRO plate reader to inject 2 mM D-luciferin (Sigma) and 10 mM ATP in the lysis buffer. Luciferase activity was expressed in relative luminescence activity.

### Purification of viral particles

HEp–2 cells were infected at high MOI in a 6-wells plate, and 24 hours p.i. cells' supernatant and cells were collected with a scraper, vortexed and clarified by centrifugation (900 g, 10 min, 4˚C). Half of the clarified supernatant was centrifuged (23, 000 rpm, 45 min, 4˚C, SW60 Ti rotor, Beckman Optima LE *80K* ultracentrifuge) through a 4 mL 30% (wt/vol) sucrose cushion in HBSS-Hepes buffer (HBSS, 25 mM Hepes, pH 7.4). Pellets were resuspended in 100 μl of HBSS-Hepes buffer and viruses were air-dried on coverslips, fixed with PFA (Paraformaldehyde; Euromedex) and then processed for IF as described below.

### Flow cytometry

HEp–2 cells were infected with RSV-mCherry at 0.5 MOI. At 22 hours p.i. cells were collected after a 5 min incubation with 1mM PBS-EDTA. Cells were incubated with a BST2 primary antibody (Proteintech; 13560-1-AP) diluted in PBS-BSA 1% (1/500) for 1 hour at 4˚C and then incubated with a secondary Alexa Fluor anti-rabbit 488 antibody diluted in PBS-BSA 1% for 1 hour at 4˚C. Lastly, the cells were fixed with PBS-PFA 4% for 20 minutes at room temperature. Samples were passed through the Fortessa BD 16 colors cytometer and the acquired data were analyzed with the FlowJo software. Two populations (infected/uninfected) were identified in infected samples based on the red fluorescence (mCherry) emitted only by the infected cells. The geometric mean intensity (GMI) of the green fluorescence (Alexa fluor 488) emitted by both populations was calculated by the software. GMI raw data were normalized to the uninfected population (raw data/mean GMI of uninfected populations from 3 samples per experiment). Normalized GMI data from 3 independent experiments were used for statistical analysis (Welch's t-test).

### Immunofluorescence

HEp–2 cells seeded and infected on coverslips were fixed with 4% PFA for 30 minutes and permeabilized for 10 minutes with 0.1% Triton-100X (Thermo Fisher; 10671652). Incubation with primary antibodies was performed for 1 hour in PBS-BSA 1%. The primary antibodies used were as follows: BST2 (Proteintech; 13560-1-AP, 1/500), TGN46 (Bio-Rad; AHP500G, 1/200), CD63 (BD transduction; 556019, 1/1000), N RSV protein (Abcam; B023, 1/1000), G RSV (Sigma; MAB858-2, 1/1000), GM130 (BD Transduction; 610822, 1/500), Giantin (Abcam; ab80864, 1/500), HA (Biolegend; clone 16B12, 1/500). Incubation with Alexa Fluor secondary antibodies (Thermo Fisher) was performed during 30 minutes along with Hoechst 33342 staining. Coverslips were mounted using Prolong Diamond Antifade Mountant (Thermo Fisher). Images were captured by a WLL Leica SP8 microscope.

Signal of BST2, detected by Immunofluorescence, presented in Figs 3 and 7 was quantified in 2 independent experiments with 30 cells analyzed per condition per experiment. Fixed cells, infected with RSVmCherry or RSVmCherryΔNS1, were labeled with phalloidin 390 (membrane) dye and with GM130 (golgi) and BTS2 antibodies. Images were acquired on a TCS SP8 confocal microscope (Leica Microsystems) using a 63X/1.4 HC PL APO CS2 Oil objective and a 4x zoom (pixel size = 0.09 μm). The entire volume of each cell was captured by performing a Z-stack acquisition (Zstep = 0.35 μm). Fluorescence emitted by the 3 labels was acquired in 3 distinct channels. Using the Imaris software Surface module (Oxford Instrument), we have performed 3D surface reconstruction in Ph390 and GM130 channels. Those volumes (ie.

membrane and golgi) were used as masks in the BST2 channel to measure fluorescence intensity. The ratio of the mean BST2 intensity signal at the plasma membrane was normalized to the total detected signal (plasma membrane + TGN). Analysis was performed on the Imaris Software. The level of colocalization was determined by acquiring confocal images from ∼ 30 cells per condition. Pearson's coefficient was then determined using the ImageJ (Fiji) software.

## Protein extraction, immunoprecipitation and Western blot

Total protein extracts were performed by adding 300μl of 2X Laemmli Buffer per 1 million cells and sonicating. Western blot experiments were carried out as follows: 20 μl of total protein extract were migrated in 12% SDS-PAGE (Bio-Rad) transferred onto PVDF membranes (Bio-Rad; 10026933), using the Trans-Blot Turbo transfer system, and incubated with corresponding antibodies in 5% milk. The primary antibodies used are the following: BST2 (Proteintech; 13560-1-AP, 1/500), RSV N (1/10000) [67], RSV M (1/1000) [67], Bip (BD Transduction Laboratories; 610978, 1/500), F (Abcam, ab43812, 1/200).

Horseradish peroxidase-coupled secondary antibodies were obtained from Promega. Membranes were revealed by chemiluminescence (Advansta; kit K-12043-D20). Quantification was performed on 3 independent experiments using Bio-Rad's Image Lab software (SD).

Immunoprecipitation experiments were performed as follows: Twenty hours post transfection cells were collected after 10 minutes of incubation in 1 mM PBS-EDTA and concentrated by centrifugation (500 g, 5 min, RT). Cells underwent lysis in the prepared buffer (50 mM Tris HCL pH 7.4, 150 mM NaCl, 0.1% SDS, 0.5% sodium deoxycholate, 1% NP40, 200 μM sodium orthovanadate, Protease inhibitor cocktail) for 1 hour at 4˚C. Immunoprecipitations were performed by incubating indicated whole cell extracts overnight at 4˚C with an anti-BST2 antibody (Tebu-bio, 157H00000684-M15), or mouse IgG CTRL (BioLegend, 40032) coupled to Dynabeads protein G (Life Technologies). The beads were washed 4 times with lysis buffer, and proteins were deglycosylated by an 1h treatment with PNGase F (NEB) then eluted in 2X Laemmli.

For the BST2 ubiquitination assay, the lysis buffer was supplemented with 20 mM *N*-ethylmaleimide (Calbiochem). BST2 Immunoprecipitation was performed as described above. The primary antibody used to reveal ubiquitin is FK2-HRP (Enzo Life Science, BML-PW0150, 1/1000) in 1% (w/v) BSA. This antibody recognizes either mono- and poly-ubiquitinylated conjugates. Membrane was incubated in advance in blocking solution (TBS–0.1% (v/v) Tween 20 supplemented with 1% (w/v) BSA).

## RT-qPCR

Cellular and viral mRNA were extracted using the QIAamp viral RNA extraction kit (Qiagen). Reverse transcription was performed with the SuperScript IV VILO kit (Thermo Fisher) and qPCR with the DyNAmo ColorFlash SYBR Green qPCR kit (Thermo Fisher). All experiments were realized according to the kit's manufacturer instructions. Data were analyzed using the $2^{-\Delta\Delta Ct}$ method. Primer sequences are available on S3 Table.

## HIV-1 release assay

Subconfluent HeLa cells were transfected with 500ng of HIV-1 NL4.3 WT or Vpu-deleted (Udel) proviral plasmid in combination with empty vector (pCi) or expression vector (50 or 100 ng) encoding Vpu-GFP or Flag-NS1 or Flag-NS2 or both using lipofectamine LTX (Invitrogen). Viral supernatant and cell lysates were harvested 48h post infection. Supernatants were collected, centrifuged 5 min at 500g, 0.45 μm-filtered and used for HIV-1 CAp24 quantification by ELISA (released CAp24) (Perkin Elmer). Viral particles released into the

supernatant were pelleted through a 20% sucrose cushion by ultracentrifugation at 150,000g for 60min and resuspended in laemmli sample buffer. Equal volumes of pelleted viruses were analyzed by western blot using mouse anti-CAp24. Cell lysates were analyzed by western blotting.

## Statistical analysis

Non parametric unpaired *t*-tests were performed on the Prism 9.5.0 Software, analyzing biological triplicates for every experiment unless indicated otherwise. **** $p < 0.0001$, *** $p < 0.001$, ** $p < 0.01$, * $p < 0.05$, ns $p > 0.05$.

## Supporting information

**S1 Fig. BST2 downregulation did not affect viral proteins expression 22h post infection with the RSV virus.** HEp-2 cells were infected with RSV virus at MOI of 1. Then, 12, 18 and 22h post infection, protein extracts were realized and processed for WB with RSV and actin antibodies
(TIF)

**S2 Fig. BST2 overexpression decreased RSV multiplication.** HEp–2 cells overexpressing HA-BST2 or HA were infected with RSV-Luc at MOI of 0,01. Then 48h post infection a luciferase assay was performed and data were expressed as a ratio of HA-BST2 cells on HA cells. 3 independent experiments with triplicates were performed and a nested t-test were performed **** $p < 0.0001$
(TIF)

**S3 Fig. Localization of TfR-HA in infected cells overexpressing TfR protein.** (A) HEp–2 cells overexpressing BST2 were infected or not with RSV-GFP-N. At 24h p.i. cells were fixed and stained with antibody against HA (red in merge). The GFP-N protein is visualized through its spontaneous green fluorescence (green in merge). Nucleus staining is shown in blue (merge). Representative images from 3 independent experiments are shown. NI (not infected). Zoom images of the red square are shown. Scale bar 10 μm.
(TIF)

**S4 Fig. FACS histogram from Fig 3A.** One representative FACS histogram of the bar graph from the Fig 3A. Non infected cells are depicted in grey and infected cells in blue.
(TIF)

**S5 Fig. Effect of infection on the cell surface level of an unrelated surface protein.** HEp–2 cells were infected or not with RSVmCherry wt. At 22h pi cells were fixed and stained with an antibody against MHCI (without permeabilization, dilution 1/100 [68]). The Geometric Mean Intensity (GMI) of MHCI at the surface of infected or not infected cells was determined by Flow Cytometry. Biological triplicates were performed for each condition. Data from 3 independent experiments.
(TIF)

**S6 Fig. Western blot quantification of Fig 4B.** The relative quantity of BST2 protein was calculated using the Biorad Image Lab Software normalized to the protein quantity of the infected condition. Data from 3 independent experiments.
(TIF)

**S7 Fig. Western blot quantification of Fig 4C.** The relative quantity of BST2 protein was calculated using the Biorad Image Lab Software normalized to the protein quantity of the infected

condition treated with the lysosomal degradation inhibitor Bafilomycin A1 or with DMSO. Data from 3 independent experiments.
(TIF)

**S8 Fig. Western blot quantification of Fig 4D.** The relative quantity of BST2 protein was calculated using the Biorad Image Lab Software normalized to the protein quantity of the infected condition treated with the proteasomal degradation inhibitor MG-132 or with DMSO. Data from 3 independent experiments.
(TIF)

**S9 Fig. Co-localization of CD63/TGN46 and BST2 in RSV P-BFP infected cells.** Confocal fluorescence microscopy of HEp-2 cells infected with RSV P-BFP at a MOI of 0,5 for 24hrs following total staining with mouse anti-CD63, rabbit anti-BST2 and sheep anti-TGN46 antibodies. Scale Bar: 20µM. Quantification of BST2 and CD63 or BST2 and TGN46 colocalization, Mean ± SD, $n$ = 1 experiment, Pearson's coefficient was measured in $\sim$20 cells per condition, and statistical analysis was done using Unpaired t test; **, P $\leq$ 0.01, ***, P $\leq$ 0.0001.
(TIF)

**S10 Fig. BST2 plasma membrane expression in cells infected with RSVΔG virus.** HEp-2 cells were infected or not with RSV and RSVΔG virus. At 24h p.i. cells were fixed and stained against BST2 (red in merge), G and RSV proteins (green in merge). Hoechst 33342 staining is shown in blue (merge). NI (not infected). Scale bar 7 µm.
(TIF)

**S1 Table. Ct Value from Figs 4A and 7D.** Ct Value (mean of triplicate) for 3 independent experiments.
(TIF)

**S2 Table. Average of viral production corresponding to Fig 7C.**
(TIF)

**S3 Table. List of primers used for RT-qPCR experiments.**
(TIF)

## Acknowledgments

Image acquisition/ cytometry experiments and analysis were performed on CYMAGES facility, part of UVSQ-Paris Saclay University. We especially thank Aude Jobart-Malfait (from the CYMAGES facility) for helpful discussions. We are grateful to Stéphane Frémont for sharing protocols, reagents and for helpful discussions. We thank Monika Bajorek for sharing NS1 plasmids. We acknowledge Aurore Desquesnes for technical support and Cédric Diot, Vincent Rincheval for helpful discussions.

## Author Contributions

**Conceptualization:** Katherine Marougka, Delphine Judith, Clarisse Berlioz-Torrent, Marie-Anne Rameix-Welti, Delphine Sitterlin.

**Data curation:** Clarisse Berlioz-Torrent, Marie-Anne Rameix-Welti, Delphine Sitterlin.

**Formal analysis:** Katherine Marougka, Delphine Judith, Tristan Jaouen, Sabine Blouquit-Laye, Gina Cosentino, Clarisse Berlioz-Torrent, Delphine Sitterlin.

**Funding acquisition:** Marie-Anne Rameix-Welti.

**Investigation:** Katherine Marougka, Delphine Judith, Tristan Jaouen, Sabine Blouquit-Laye, Gina Cosentino, Delphine Sitterlin.

**Methodology:** Katherine Marougka, Delphine Judith, Clarisse Berlioz-Torrent, Marie-Anne Rameix-Welti, Delphine Sitterlin.

**Supervision:** Clarisse Berlioz-Torrent, Delphine Sitterlin.

**Validation:** Clarisse Berlioz-Torrent, Marie-Anne Rameix-Welti, Delphine Sitterlin.

**Visualization:** Katherine Marougka, Delphine Judith, Clarisse Berlioz-Torrent, Delphine Sitterlin.

**Writing – original draft:** Katherine Marougka, Delphine Judith, Clarisse Berlioz-Torrent, Marie-Anne Rameix-Welti, Delphine Sitterlin.

**Writing – review & editing:** Katherine Marougka, Delphine Judith, Clarisse Berlioz-Torrent, Marie-Anne Rameix-Welti, Delphine Sitterlin.

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
