## [Decision Letter · Decision Letter 0]

17 Dec 2023

Dear MC sitterlin,

Thank you very much for submitting your manuscript "BST2/Tetherin, a new restriction factor of respiratory syncytial virus, is antagonized by viral NS1 protein." for consideration at PLOS Pathogens. As with all papers reviewed by the journal, your manuscript was reviewed by members of the editorial board and by several independent reviewers. In light of the reviews (below this email), we would like to invite the resubmission of a significantly-revised version that takes into account the reviewers' comments.

We cannot make any decision about publication until we have seen the revised manuscript and your response to the reviewers' comments. Your revised manuscript is also likely to be sent to reviewers for further evaluation.

Sincerely,

Alexander Bukreyev, Ph.D.

Academic Editor

PLOS Pathogens

Kanta Subbarao

Section Editor

PLOS Pathogens

Kasturi Haldar

Editor-in-Chief

PLOS Pathogens

orcid.org/0000-0001-5065-158X

Michael Malim

Editor-in-Chief

PLOS Pathogens

orcid.org/0000-0002-7699-2064

Reviewer's Responses to Questions

**Part I - Summary**

Reviewer #1: In the current study the authors look at the impact of the anti-viral restriction factor BST2 on RSV infection in vitro. it is an interesting paper but more work is needed to confirm the mechanism that RSV evades the BST response and that it is a direct NS1 effect.

Reviewer #2: The manuscript titled "BST2/Tetherin, a new restriction factor of respiratory syncytial virus 1, is antagonized by the viral NS1 protein" is a research article describing the interaction of BST2/Tetherin in RSV infections in vitro. The authors also explored a potential evasive effect of RSV on BST2 induced by the viral protein NS1. While the authors presented convincing evidence of BST2's interaction with RSV, the demonstration of the antagonistic effect by NS1 requires further investigation. The paper has the potential to advance the knowledge of RSV biology. However, in its current form, the manuscript is too preliminary for acceptance. The reasons for rejection and suggestions to improve the manuscript quality are outlined below.

Reviewer #3: Marougka and colleagues investigated whether the cellular restriction factor tetherin/BST2 Is active against RSV. In brief, they show that siRNA-mediated knock-down of tetherin slightly increases RSV titers in a cell line at a late stage of infection. Further, an interaction of tetherin with components of the viral envelope is demonstrated and it is shown that RSV downregulates tetherin expression in a NS1 dependent fashion and that a delta NS1 virus is more sensitive to tetherin’s antiviral activity than the WT virus. The findings are of interest to the field. However, important points remain open.

**Part II – Major Issues: Key Experiments Required for Acceptance**

Reviewer #1: Major comments:

1. How do you separate out the broader anti-IFN role of NS1 from the specific anti-BST2 role? NS1 antagonises IFN, which would increase BST2, in its absence you will get more of both.

2. Need to repeat the studies with an NS2 knockout virus and change line 303 – there is NS2 protein in the pull down, it might have a role.

Reviewer #2: It is widely accepted that Interferon-induced genes (ISGs) are not highly expressed in unstimulated cells, with exceptions noted in certain types of tumors characterized by elevated ISG expression. During the viral entry process, viruses can activate Pattern Recognition Receptors (PRRs). This activation triggers a cascade culminating in the expression and secretion of type I interferons (IFN). The paracrine and autocrine activation of Interferon receptors activates the STAT family proteins resulting in the expression of ISGs.

The authors affirm that this manuscript seeks to investigate BST2's interaction with RSV under more physiological conditions than currently available in the literature. However, despite their objective to study physiological conditions, the chosen cell line for this investigation presents numerous inconsistencies and is unlikely to serve as an appropriate model. According to the ATCC website, the "HEp-2 cell line contains HeLa marker chromosomes and was derived via HeLa contamination(...)." (https://www.atcc.org/products/ccl-23). The uncertainty surrounding the cell lineage and the presence of high basal expression of BST2 create challenges in interpreting the results. I recommend that the authors redirect their efforts toward BCi-NS1.1 cells. If the authors decide to keep Hep-2 cells (possibly HeLa cells) they should authenticate their Hep-2 cell line avoiding future inconsistencies in the literature. The authors should also verify the effect of IFN stimulation and downstream effectors activation in their models, checking if the expression of BST2 in Hep-2 cells resembles the one in BCi-NS1.1 cells.

Beyond the cell-related concerns regarding this manuscript, Marougka et al. did not attempt to elucidate the mechanisms underlying BST2 downregulation by NS1 as well. For instance, the only evidence provided regarding mechanisms relies on bafilomycin and MG132-treated cells. They should corroborate their drug-based assay by measuring mono-ubiquitination and poly-ubiquitination of BST2, as well as performing confocal experiments aiming to colocalize BST2 and acidic compartments or proteasome components.

Additional comments and suggestions:

1. Fig 1B. The authors suggest that BST2 depletion improves viral production. Fig 1B shows a multi-step replication curve of RSV. The authors called the difference found modest but highly reproducible. RSV can present a very long replication cycle and 48 hours may not reflect a truly multistep curve. I advise the authors to define the time required for a one-cycle curve in the cell lineage they are using and perform a multicycle curve based on the one-cycle curve timeframe.

2. In Fig 3B, the authors quantified the amount of BST2 in the plasma membrane, however, no plasma membrane marker was used. The use of phalloidin to reconstruct the cellular plasma membrane may not generate an accurate location of the plasma membrane. If the authors think that there are no better options, the authors should describe better how the plasma membrane reconstruction and BST2 localization/quantification were performed. Additionally, the authors should clarify why the total amount of BST2 is based on the presence of BST2 in TGN (along with the plasma membrane) instead of the total amount of BST2 present in the whole cell.

3. In lines 222-224, the authors affirm that "We showed in Fig 3D that BST2 mRNA level was not significantly modified in infected cells compared to non-infected cells, suggesting that the regulation of the amount of BST2 doesn’t take place at the transcriptional level." The use of relative quantification in their analysis avoids the detection of global expression down-regulation, which is common for many virus families. Caution should be exercised in this statement, and some other method should corroborate such conclusion. Additionally, providing the primer sequences and the reference gene used for the relative quantification would enhance the transparency and reproducibility of the study.

4. Comparing Hep-2 cells (possibly HeLa cells) and BCi NS1.1 cells is problematic. The former expresses a high level of ISGs in the absence of IFN activation, while the latter appears to depend on virus infection and likely IFN stimulus. These are entirely different biological models and should not be directly compared unless experimental evidence proves otherwise.

5. Marougka et al also failed to quantify the expression of BST2 mRNA in the experiments with RSVΔNS1, in the assays using BCi NS1.1 cells, and in the experiments using plasmid transfections. The quantification of BST2 mRNA should be part of all experiments comparing BST2 variations.

6. Bafilomycin inhibits endosome acidification, and MG132 inhibits proteasome-mediated degradation. Although inhibitory drugs can serve as initial evidence for biological phenomena, the effect is not specific. Drugs frequently affect multiple processes inside the cells and a more specific approach should be used to corroborate the data using drugs.

7. The conclusion that BST2 degradation upon infection is dependent on the endolysosomal pathway and ubiquitin is not supported by the manuscript data. MG132 inhibits proteasome-mediated degradation but does not directly influence ubiquitination.

8. The authors' statement “an RSVΔNS1 virus is more sensitive to BST2 restriction than the wild-type virus because it fails to decrease BST2 cell surface expression” is supported by comparing Fig 6A’s bottom panel and middle panel. However, the bottom panel in Fig 6A resembles the same pattern observed in the bottom panel in Fig 3B. The authors should discuss why RSV infection in Fig 3B resembles RSVΔNS1 infection in Fig 6A.

Reviewer #3: The effect of tetherin knock-down on RSV infection is minor. Confirmation of antiviral activity is needed under conditions of overexpression. Thus, infection of cell lines overexpressing tetherin relative to control cell lines must be analyzed. Further, it would be more convincing to show the average of several independent experiments in figure 1B as compared to the results of a representative experiment.

It is important to investigate whether NS1 alone like Vpu is able to antagonize tetherin. This question can be addressed in a straightforward manner by analyzing whether NS1 expression rescues release of retroviral particles from inhibition by tetherin.

Figure 6C: It is important to state the viral titers in the figure legend. If delta NS1 virus grew to much lower titers as compared to WT virus then the data are not very telling.

**Part III – Minor Issues: Editorial and Data Presentation Modifications**

Reviewer #1: Minor comments:

1. Why not take the in vitro assay out longer in Fig 1B? The effect only apparent after 48 hrs, can you look at 72?

2. In 2A: the F blot, what is the higher band that is present in all the lanes?

3. Why is there BST2 in the HeLa NI images but not the primary cells? Does that affect interpretation?

Reviewer #2: Minor issues

1) Fig 4, if the monolayer was infected in a high MOI, why just a small fraction of cells is infected?

2) Flow cytometry histograms should be displayed in the figures.

3) RSVΔNS1 should be added to Fig 1B.

4) Fig 1B legend: What does “Representative data of 3 independent experiments are shown” mean? Does it mean that only data from one experiment was used in the graph? If that is the case, the authors should add the other experiments to the graph as well.

5) In Line 219, the authors should add the Figure number to the flow cytometry results they mentioned.

6) In Fig 6A, cells in the bottom panel seem smaller than the middle and top panels. Could the authors verify if the magnification used is the same?

7) Lines 388-389: the authors mentioned that BCi-NS1.1 expression of BST2 is dependent on interferon secretion. Interferon detection was not performed in any of the experiments described in this manuscript.

8) The authors should disclose enough information in the Material and Methods session to allow the reproducibility of the study. Information like antibody dilutions, primer sequences, confocal settings, and details in the image analysis are missing.

9) Line 521-527. Is the BST2 quantification performed per z-stack? A better description of the image analysis should be added to the manuscript.

Reviewer #3: It is a significant weakness of the study that the impact of tetherin on RSV spread in primary lung cells and the contribution of tetherin to inhibition of RSV infection by IFN were not analyzed. This should be clearly stated in the discussion section.

PLOS authors have the option to publish the peer review history of their article (what does this mean?). If published, this will include your full peer review and any attached files.

Reviewer #1: No

Reviewer #2: No

Reviewer #3: No
---

## [Decision Letter · Decision Letter 1]

12 Aug 2024

Dear MC sitterlin,

Thank you very much for submitting your manuscript "Antagonism of BST2/Tetherin, a new restriction factor of respiratory syncytial virus, requires the viral NS1 protein." for consideration at PLOS Pathogens. As with all papers reviewed by the journal, your manuscript was reviewed by members of the editorial board and by several independent reviewers. The reviewers appreciated the attention to an important topic. Based on the reviews, we are likely to accept this manuscript for publication, providing that you modify the manuscript according to the review recommendations. Specifically, please address the comments of the Reviewer #2 and Reviewer #3 to the degree you can without performing new experiments. 

Sincerely,

Alexander Bukreyev, Ph.D.

Academic Editor

PLOS Pathogens

Kanta Subbarao

Section Editor

PLOS Pathogens

Michael Malim

Editor-in-Chief

PLOS Pathogens

orcid.org/0000-0002-7699-2064

Reviewer Comments (if any, and for reference):

Reviewer's Responses to Questions

**Part I - Summary**

Reviewer #1: The authors have addressed my concerns

Reviewer #2: The manuscript titled "BST2/Tetherin, a New Restriction Factor of Respiratory Syncytial Virus 1, is Antagonized by the Viral NS1 Protein" describes the interaction of BST2/Tetherin in RSV infections in vitro. In my initial review, I recommended the rejection of the paper due to its preliminary nature. While the revised version shows significant improvement, it remains fundamentally preliminary. Therefore, my recommendation remains rejection.

some of Marougka et al.'s conclusions appear somewhat rushed. For instance, they demonstrated that the ubiquitination of BST-2 increases in RSV-infected cells. This is interesting data; however, they fail to clarify how they differentiate between polyubiquitination (a signal for proteasome degradation) and monoubiquitination (a signal for endosomal degradation).

My comments and suggestions are outlined below.

Reviewer #3: The authors have assessed the effect of exogenous tetherin on RSV infection and now present the average of several experiments assessing the effect of tetherin on relative production of viral particles. This is appreciated and makes the manuscript stronger. However, two important points remain insufficiently addressed:

**Part II – Major Issues: Key Experiments Required for Acceptance**

Reviewer #1: N/A thank you for softening the conclusion about NS1

Reviewer #2: In the current version, the authors have convincingly proved that 1) BST-2 is a restrictive factor for RSV infections; 2) BST-2 is present in the plasma membrane of uninfected cells; 3) when infected by RSV, the BST-2 migrates from the plasma membrane to acidic compartments; 4) the downregulation of BST-2 in Hep-2 cells infected by RSV is not at the transcriptional level; 5) RSV infection induces BST-2 ubiquitination in Hep-2 infected cells; 6) RSV NS1 protein immunoprecipitated together with BST-2 in Hep-2 infected cells; and 7) BST-2 in RSV ΔNS1 infected Hep-2 cells tends to stay at the plasma membrane when compared to wt RSV.

Some of Marougka et al.'s conclusions seem somewhat rushed. For instance, they demonstrated that the ubiquitination of BST-2 increases in RSV-infected cells and concluded that BST-2 is polyubiquitinated in these cells. While this is intriguing data, general ubiquitination staining cannot distinguish between polyubiquitination (a signal for proteasome degradation) and monoubiquitination (a signal for endosomal degradation). If the authors have differentiated mono and polyubiquitination in this manuscript is not clear how. They should clarify this issue by giving a better description in the material and method session.

BST-2 is described as a dimer, with the C-terminal GPI anchor trapping viral membranes at the plasma membrane. The authors demonstrated RSV trapped at the membrane in Figure 1, indicating that RSV is recognized by BST-2 in Hep-2 infected cells. If the authors' hypothesis is correct, RSV would escape being trapped by BST-2 at the plasma membrane, and RSV-ΔNS1 would accumulate at the plasma membrane. Figure S9 shows RSV colocalizing with BST-2 in areas close to the plasma membrane. In my opinion, this indicates that wt RSV cannot efficiently escape from BST-2. The authors did not show the same staining as Figure S9 for RSV-ΔNS1 for comparison purpose.

After being activated by viruses membranes, the complex BST-2-virus is endocytosed and the virus is inactivated at the acidic compartments. Part of the endocytosed BST-2 is recycled to the plasma membrane through the TGN. Therefore, the colocalization of BST-2 with TGN showed in Marougka et al does not necessarily mean that BST-2 is being targeted for destruction and may be the natural course of BST-2 after activated by viruses.

Viruses have evolved ways to evade BST-2 binding. The most studied strategy for BST-2 evasion is HIV’s Vpu. Vpu seems to induce the degradation of BST-2 by proteasomes and inhibition of BST-2 recycling from TGN. The authors claim that a similar mechanism is observed for RSV NS1. In my opinion, this manuscript does not produce enough evidence to support their claim.

Other major questions are listed below:

1. Inhibition of degradation pathways by MG-132 or Bafilomycin is expected to increase the concentration of most proteins in the cells, as proteins targeted for destruction will remain in the cell. For this reason, drug data needs to be corroborated with additional data before drawing any conclusions regarding BST-2 degradation pathways.

2. In Figure 3, the authors did not provide a detailed method for quantifying BST-2 in the plasma membrane and whole cell. The absence of a specific plasma membrane antibody is a major concern. This is particularly important in RSV-infected cells, where BST-2 staining is widely distributed.

Reviewer #3: PFU/ml are now provided in table S2. These results show that the effect of tetherin on virus production was analyzed under conditions resulting in a 2 log difference between WT RSV and Delta NS1 RSV titers. One could speculate that this pronounced difference in titers accounts for the differential tetherin sensitivity - thus it is conceivable that a mutant virus that replicates with markedly reduced efficiency as compared to the WT virus is also generally more sensitivity to inhibition by ISGs.

The data with vpu defective HIV-1 indicate that NS1 is not a tetherin antagonist. This is important information and should be shown in the manuscript. At the very least, the results should be stated and discussed in the manuscript.

**Part III – Minor Issues: Editorial and Data Presentation Modifications**

Reviewer #1: None

Reviewer #2: Minor issues:

- The authors’ one-step curve shown that the RSV cycle is around 24 hours in duration. Therefore, their multi-step curve has only two full cycles. More timepoints would be beneficial for Figure 1 clarity.

- Figure S5 shows a 2.4-fold decrease in the total amount of BST-2 in Hep-2 cells infected by RSV. However, confocal images suggest that BST-2 migrates from the plasma membrane to the TGN. Upon visual inspection of the confocal images, it appears that the total amount of BST-2 remains constant, with only the location changing. The authors should verify if the time after infection is consistent between the confocal experiment and the Western Blot. Based on Figure 3, I would expect that at 24 hours post-infection, the total amount of BST-2 would be similar in non-infected and RSV-infected Hep-2 cells.

- In my initial review, I did not mean to replace the bar graph with the FACS histogram (Figure 3A). I meant that the FACS histogram should be added to the figure along with the bar graph (or as supplemental figure).

Reviewer #3: (No Response)

PLOS authors have the option to publish the peer review history of their article (what does this mean?). If published, this will include your full peer review and any attached files.

Reviewer #1: No

Reviewer #2: No

Reviewer #3: No

Figure Files:

Data Requirements:

Reproducibility:

References:

---

## [Editor Report · Decision Letter 2]

22 Oct 2024

Dear MC sitterlin,

We are pleased to inform you that your manuscript 'Antagonism of BST2/Tetherin, a new restriction factor of respiratory syncytial virus, requires the viral NS1 protein.' has been provisionally accepted for publication in PLOS Pathogens.

Best regards,

Alexander Bukreyev, Ph.D.

Academic Editor

PLOS Pathogens

Kanta Subbarao

Section Editor

PLOS Pathogens

Michael Malim

Editor-in-Chief

PLOS Pathogens

orcid.org/0000-0002-7699-2064
---

## [Editor Report · Acceptance letter]

12 Nov 2024

Dear MC sitterlin,

We are delighted to inform you that your manuscript, "Antagonism of BST2/Tetherin, a new restriction factor of respiratory syncytial virus, requires the viral NS1 protein.," has been formally accepted for publication in PLOS Pathogens.

Best regards,

Michael Malim

Editor-in-Chief

PLOS Pathogens

orcid.org/0000-0002-7699-2064